# Lifting the veil on the dynamics of neuronal activities evoked by transcranial magnetic stimulation

**Bingshuo Li[1,2,3,4,5], Juha P Virtanen[1,2,3,4], Axel Oeltermann[6], Cornelius Schwarz[1,3], Martin A Giese[2,3], Ulf Ziemann[4], Alia Benali[1,2,3]\***

[1]Systems Neurophysiology, Werner Reichardt Centre for Integrative Neuroscience, University of Tübingen, Tübingen, Germany; [2]Section on Computational Sensomotorics, Werner Reichardt Centre for Integrative Neuroscience, University of Tübingen, Tübingen, Germany; [3]Department of Cognitive Neurology, Hertie Institute for Clinical Brain Research, University of Tübingen, Tübingen, Germany; [4]Department of Neurology and Stroke, Hertie Institute for Clinical Brain Research, University of Tübingen, Tübingen, Germany; [5]Graduate Training Centre/ International Max Planck Research School for Cognitive and Systems Neuroscience, University of Tübingen, Tübingen, Germany; [6]Max Planck Institute for Biological Cybernetics, Tübingen, Germany

**Abstract** Transcranial magnetic stimulation (TMS) is a widely used non-invasive tool to study and modulate human brain functions. However, TMS-evoked activity of individual neurons has remained largely inaccessible due to the large TMS-induced electromagnetic fields. Here, we present a general method providing direct in vivo electrophysiological access to TMS-evoked neuronal activity 0.8–1 ms after TMS onset. We translated human single-pulse TMS to rodents and unveiled time-grained evoked activities of motor cortex layer V neurons that show high-frequency spiking within the first 6 ms depending on TMS-induced current orientation and a multiphasic spike-rhythm alternating between excitation and inhibition in the 6–300 ms epoch, all of which can be linked to various human TMS responses recorded at the level of spinal cord and muscles. The advance here facilitates a new level of insight into the TMS-brain interaction that is vital for developing this non-invasive tool to purposefully explore and effectively treat the human brain.
DOI: https://doi.org/10.7554/eLife.30552.001

**\*For correspondence:** alia.benali@uni-tuebingen.de

**Competing interests:** The authors declare that no competing interests exist.

## Introduction

In 1985, when *Barker et al. (1985)* applied a pulsed magnetic field to a subject's head to selectively evoke responses in a hand muscle, the scientific community was fascinated by this pain-free, electro-deless, and non-invasive tool for brain stimulation. Since then, transcranial magnetic stimulation (TMS) has attracted enormous interest in neurology, psychiatry, and applied neuroscience research for its unique capability of non-invasively activating neuronal populations and inducing plasticity. Despite its impressive array of applications and fast-growing popularity (*Cohen et al., 1998*; *Walsh and Cowey, 2000*; *Reis et al., 2008*; *Funke and Benali, 2011*; *Lefaucheur et al., 2014*), TMS is poorly understood physiologically as we know very little about how TMS interacts with the brain at the level of neurons and circuitries. Although various sources of indirect evidence obtained from human studies support the use of TMS in multiple contexts (*Di Lazzaro et al., 2008*; *Chung et al., 2015*; *Suppa et al., 2016*), our very limited insight into the neurophysiology of TMS

**eLife digest** Being able to tap into someone's brain activity by holding loops of wires above their head sounds a little like the stuff of science fiction. And yet this technique, known as transcranial magnetic stimulation or TMS, is used in research and to treat many brain disorders. TMS emits a pulsed magnetic field that induces tiny electrical currents in the underlying brain tissue, activating that region of the brain. But exactly how these currents affect the individual neurons and networks within activated brain regions remains unclear.

The main reason for this is that we cannot use conventional electrode-based techniques to study neuronal activity during TMS because its strong electromagnetic interferences mask the signals from the electrodes. Several groups have found ways to overcome this problem. However, their methods are technically demanding and specific to one single animal model –limitations that could present an obstacle for many laboratories. Li et al. therefore set out to develop a simple and widely accessible method to study neuronal activities under TMS.

The resulting method makes it possible to measure the activity of individual neurons roughly 1/1,000th of a second after applying TMS. To show that the technique works, Li et al. induced small movements in the forelimbs of rats by applying TMS to the brain region that controls the forelimbs, while measuring the activity of neurons at the same time. This revealed, for the first time, how the neurons responsible for the forelimb movements responded to TMS. The observed TMS-triggered neuronal activity continued long after the TMS pulse had ended. The activity also varied depending on the direction of TMS-induced currents in the brain.

This new method opens up the possibility to conveniently study – in rodents or other animals – how TMS procedures that are used in patients affect neuronal activity. Li et al. hope this will make it easier to develop, study and refine these procedures, and lead to advances in TMS therapies.

DOI: https://doi.org/10.7554/eLife.30552.002

remains a bottleneck that hampers the utilization and the development of TMS applications, blocking the exciting potential of this non-invasive brain stimulation tool.

One critical reason behind this inadequacy is the absence of a research platform on which TMS-evoked neuronal activities can be investigated in vivo in real time. Extracellular electrophysiology (EEP) with microelectrodes is the gold standard for studying brain activities at the level of neurons (*Scanziani and Häusser, 2009*). However, a single TMS pulse, characterized by an alternating tesla-level magnetic field with a center frequency of approximately 4 kHz (*Ilmoniemi et al., 1999*; *Wagner et al., 2007*), generates an array of strong interferences disturbing the acquisition of EEP signal. As evidenced by the pioneering works on TMS-EEP (*Moliadze et al., 2003*; *Pasley et al., 2009*), artifacts from TMS resulted in a significant amount of data loss that precluded the investigation of TMS-evoked neuronal activities for up to 100 ms after each TMS pulse. Several groups used innovative imaging methods to circumvent this problem (*Allen et al., 2007*; *Kozyrev et al., 2014*; *Murphy et al., 2016*); however, their investigations were restricted to the upper cortical layers and they were unable to reach the single-neuron and millisecond precision needed to decipher the intricate interactions between TMS and neurons. Recently, two groups reported TMS-EEP methods for non-human primate research. One of the methods utilized custom-built TMS coil and offline correction to minimize the TMS-induced data loss to a median time of 2.5 ms (*Tischler et al., 2011*), while the other used a combination of custom-built coil, amplifier modifications, field sensing, active compensation, and offline correction to minimize the data loss to 1 ms (*Mueller et al., 2014*). Despite their success in artifact reduction, these methods face a major limitation that the technical expertise required for their implementations, especially custom-building TMS coils and field sensing, is not widely accessible to the neuroscience community. More importantly, these methods were developed solely for non-human primate research, which is used for investigating the neuronal underpinnings of high-level cognitive functions and therefore is not best suited for investigations concerning basic neurophysiology on the level of cells and detailed microcircuits.

With the aim of establishing a widely applicable in vivo experimental platform to study the dynamics of TMS-evoked neuronal activities and to further develop the scientific and clinical applications of this powerful technique, we engineered a novel method for TMS-EEP that is suitable for, but

not limited to, laboratory rodents, which are widely accessible and offer a rich repertoire of experimental techniques including transgenic and optogenetic tools. The method is compatible with existing standard TMS coils and allows the recording of neuronal activities 0.8–1 ms after the onset of various types of TMS stimuli by attenuating artifacts resulting from magnetic induction, electric field coupling, and vibrations. Furthermore, the method allows for the instantaneous determination of TMS-driven inadvertent charge injection into the neural tissue, a problem that has been overlooked by almost all prior TMS-EEP studies. In the following sections, we will present this methodological advance and demonstrate its potential by unveiling neuronal activities in the layer V of the primary motor cortex (M1) that accompany cortically evoked unilateral muscle activations by monophasic single-pulse TMS (mspTMS). Additionally, we will also demonstrate that mspTMS modulates different neuronal circuits depending on the orientation of the stimulating current or the time-window of investigation.

## Results

### Attenuation of the induction artifact

The induction artifact (*Figure 1*, period indicated in blue) is created as coil-generated rapid magnetic flux change induces voltages within loops formed along an EEP recording assembly. Owing to

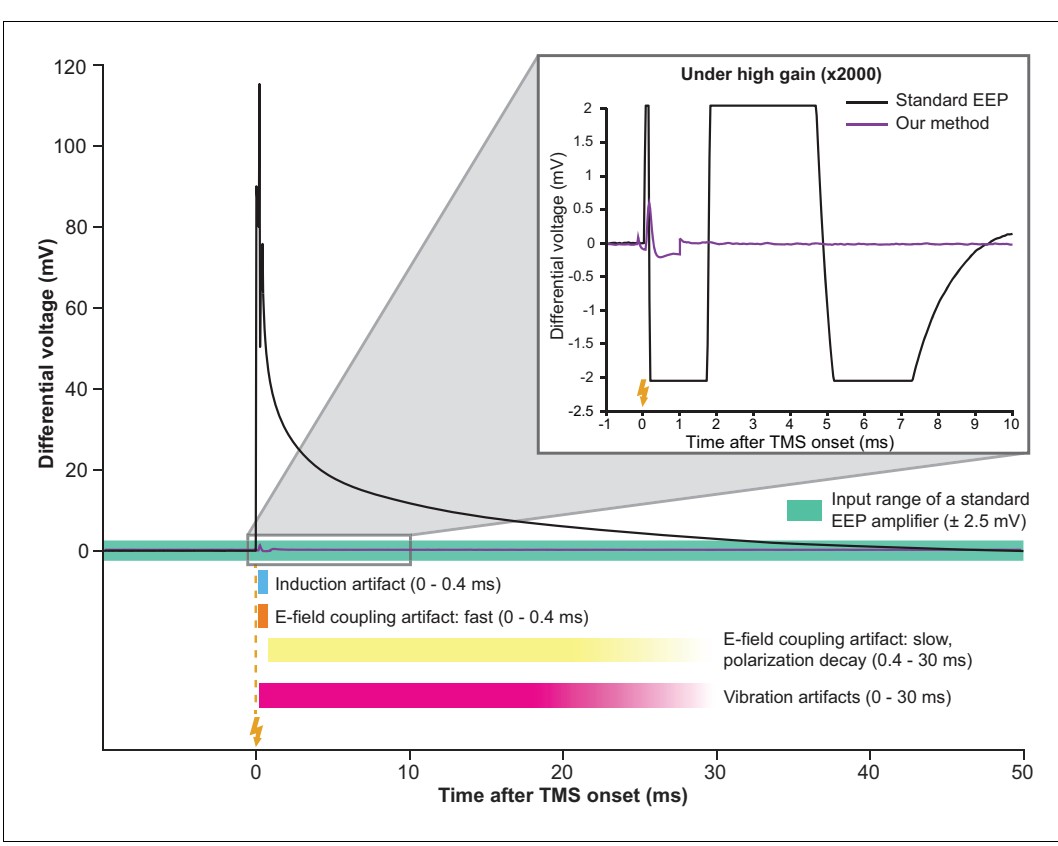

**Figure 1.** Simultaneous TMS-EEP recording requires artifact attenuation in multiple dimensions. The full waveform of a TMS artifact recorded differentially under low-gain (x4) using a high impedance amplifier. The artifact consists of a series of sharp deflections (induction and fast E-field coupling artifacts) and a long tail (polarization decay artifact resulted from E-field coupling). The slow polarization decay artifact renders the signal out of range (indicated by the green area) until ca. 30 ms post-TMS in a standard EEP system. Vibration artifacts (see *Figure 4B*) are not visible here due to low amplification. The inset shows that under high-gain (x2000) needed for EEP, TMS artifacts lead to long signal saturation in a standard EEP system (bandpass 300–5000 Hz) while producing negligible interference in our method. Lightning symbol, TMS onset (at 0 ms). E-field, electric field.
DOI: https://doi.org/10.7554/eLife.30552.003

the large rate of flux change, the induction artifact, if transmitted to the high-gain and filter stages of an EEP amplifier, can easily lead to signal saturation and data loss (*Figure 1* inset). To address this, we developed a gated multi-stage TMS-EEP amplifier (*Figure 2*). It consists of a differential pre-amplifier (Pamp) stage of gain four and a filter-amplifier (Famp) stage of gain 500, separated by an ultra-low capacitance/charge injection solid-state analog switch (SSSW) controlled by optically coupled digital signal synchronized to TMS. The components of the amplifier were chosen to provide the optimal balance between voltage and current noise with the source impedance of EEP micro-electrodes. The Pamp stage must be able to maintain its high impedance character when being perturbed by TMS. Failing to do so will result in excessive induction current in the input wires that leads to electrical stimulation of the brain. In our design, the electronic components and supply voltage of the Pamp stage were chosen so it can tolerate ±7.9 V input during TMS. Due to its high-gain and filters, the Famp stage must be protected from TMS by SSSW that grounds the input to Famp for a user-defined time interval around TMS onset (e.g. from 0.2 ms pre- to 0.8 ms post-TMS). The SSSW was strategically placed behind the input capacitor of a high-pass filter of the Famp so that the input capacitor is preconditioned to any DC bias in the microelectrodes before the end of the grounding period. To minimize ground bounce and to reduce ground loop, the external digital signal that controls SSSW is connected to the amplifier circuit through an optocoupler (OC). In addition, to protect the amplifier circuit from TMS-induced fields, the circuit board of the amplifier is mounted inside a 1.5-mm-thick grounded aluminum enclosure. The Pamp, as well as DC-DC converters, is housed inside a metal shield for additional protection. Furthermore, to minimize artifacts from vibrations due to the loud click sound of TMS coils, solder contacts, instead of spring-loaded connectors, were used whenever possible, as well as polyphenylene sulfide film capacitors, since they do not generate piezoelectric voltages from vibration. The default frequency response of the amplifier was set from 300 Hz (−3 dB) to 5 kHz (−6 dB), but the lower bound of the passband can be adjusted to 4 Hz as needed for different applications. A simplified circuit diagram of the amplifier, including the model number of its critical components, is shown in *Figure 2*.

## Attenuation of electric field coupling artifacts

The coil-emitted electric field gives rise to another type of artifacts. When a TMS pulse is triggered, a large current driven by a kV-level voltage pulse flows through the coil. Inadvertently, this process

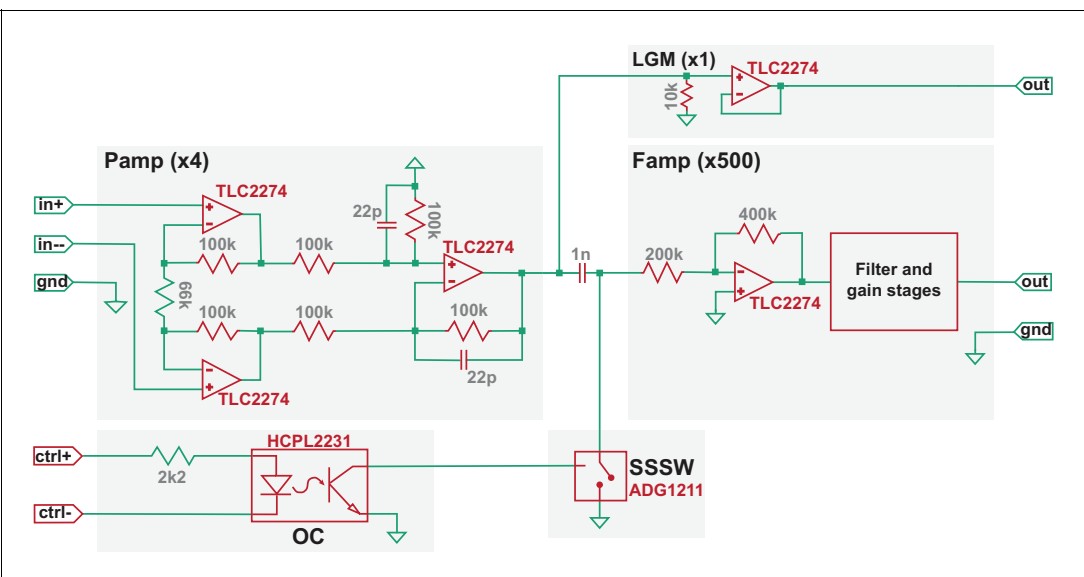

**Figure 2.** Simplified circuit diagram of the TMS-EEP amplifier. Model numbers of the most critical electronic components are noted in red, the values of certain elementary components are noted in grey (units: ohm for resistor; farad for capacitor), and the amplification factor for each stage is indicated in parenthesis. Pamp, pre-amplifier stage; Famp, filter-amplifier stage; OC, optocoupler; SSSW, solid-state analog switch; LGM, low-gain monitoring channel.

DOI: https://doi.org/10.7554/eLife.30552.004

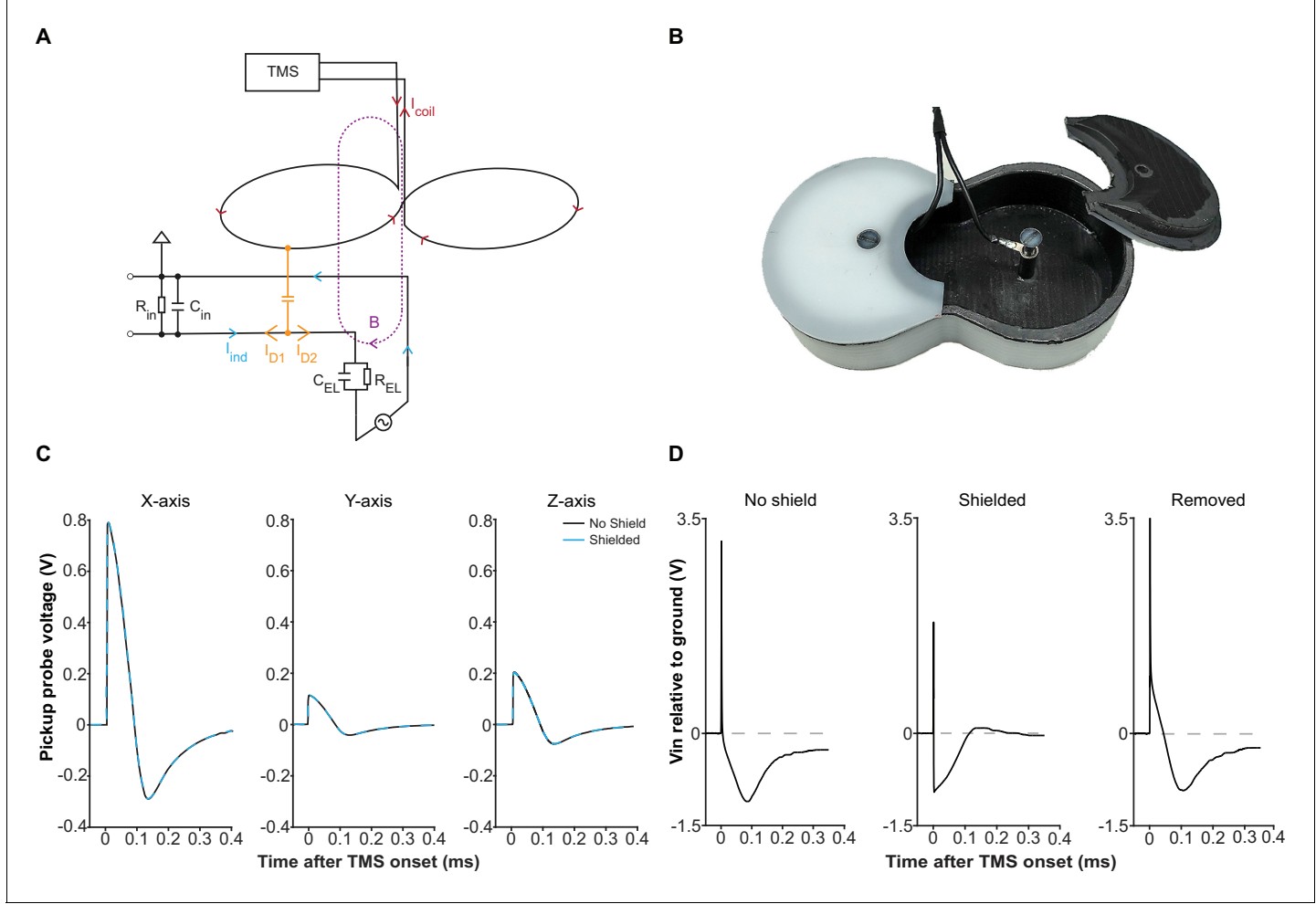

**Figure 3.** Electric field coupling in TMS-EEP and its attenuation. (A) A schematic illustrating how electric field coupling interferes with the EEP recording circuit. Here, the loop between one microelectrode and the ground is used as an example. The microelectrode is modeled as a parallel resistor-capacitor for simplification. Note how displacement current ($I_D$), generated by electric field coupling, propagates in both directions once it enters the circuit, while the magnetically induced current ($I_{ind}$) only propagates in a circular manner. The branch of displacement current ($I_{D1}$) that flows toward the input end of the amplifier opposes the $I_{ind}$, counteracting the magnetically induced voltage change across the amplifier input resistance ($R_{in}$). The other branch ($I_{D2}$) flows toward the electrode and can cause polarization at the microelectrode tip. Abbreviations: B, magnetic field; $C_{EL}$, electrode capacitance; $C_{in}$, amplifier input capacitance; $I_{coil}$, TMS coil current; $R_{EL}$, electrode resistance. (B) The electrical shield constructed for the Magstim D25 coil. The shield fits tightly with the coil and is grounded through the EEP recording system. (C) Induction waveforms from a pickup probe positioned right below the coil center, along the X-, Y- and Z- axis, with or without the shield, under mspTMS at maximum intensity. Along each axis, the waveforms obtained under shielded and no shield condition overlap, confirming that the shield does not attenuate the magnetic output of the TMS coil. (D) Input voltage to a high impedance buffer (AD825, $V_s = \pm15V$), measured with a 1.5 MΩ (1 kHz) microelectrode, and an Ag/AgCl ground electrode in normal saline under mspTMS at maximum intensity with or without the shield. Signal in the 'Removed' condition was obtained by taking the difference between the waveforms in 'No shield' and 'Shielded' condition. The shield restored the correct induction waveform and abolished the voltage offset that leads to the decay.

DOI: https://doi.org/10.7554/eLife.30552.005

emits an electric field that injects a displacement current into the EEP recording assembly through capacitive coupling (*Figure 3A*). In a short time-interval (<0.4 ms; *Figure 1*, period indicated in orange), as the coil current rapidly fluctuates, the displacement current generates a fast-changing artifact in EEP signal. In a long time-interval (tens of ms; *Figure 1*, period indicated in yellow), a decay-like artifact is observed as the displacement current polarizes the electrochemical double-layer of microelectrode tips and thereby generates a decaying waveform while the double-layer returns to its equilibrium potential. Depending on stimulation intensity, electrode impedance, and filter settings, the decay may persist with relatively high signal values for tens of milliseconds before re-

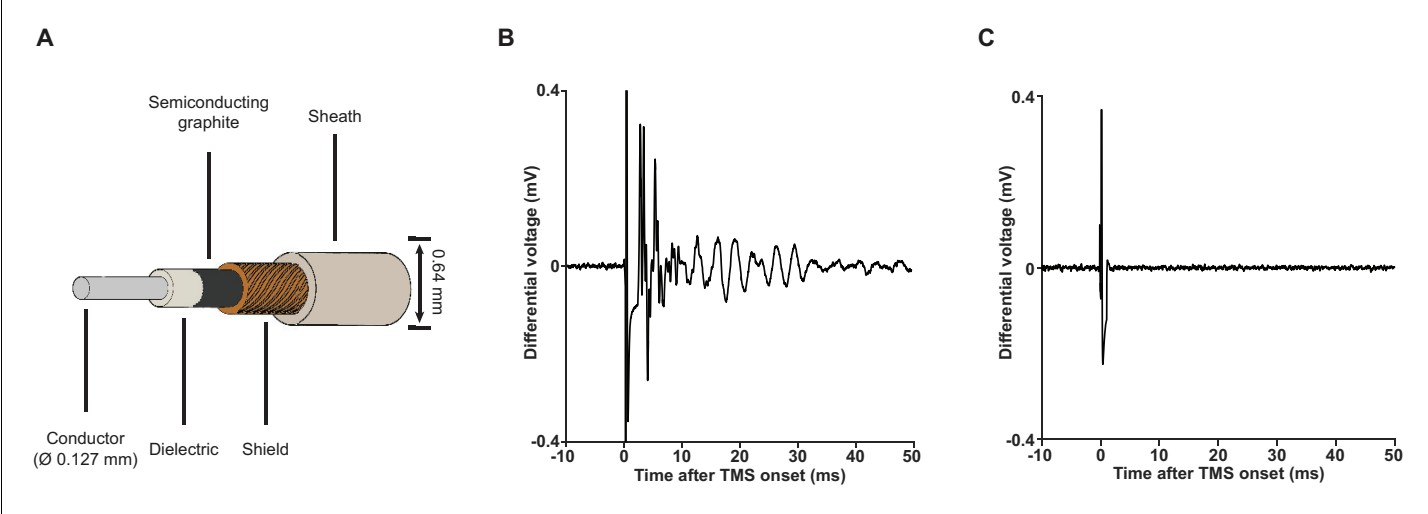

**Figure 4.** Low-noise miniature coaxial cable attenuates vibration artifacts. (**A**) A schematic illustration of the 36-gauge low-noise miniature coaxial cable. A semiconducting layer of graphite is added between the braided shield and the dielectric of the cable to drain triboelectric charges, rendering the cable insensitive to vibration. (**B**) An example of vibration artifacts recorded under the standard EEP conditions (x2000 using our TMS-EEP amplifier; 1.5 MΩ microelectrode pair; Ag/AgCl ground electrode; normal miniature coaxial cable) in a saline bath after induction and electric field coupling artifacts were suppressed. The vibration artifacts can manifest in multiple types of waveform, depending on the parts of the recording assembly that are perturbed and the resonance properties of these parts. (**C**) The implementation of low-noise miniature coaxial cables attenuated the vibrational artifacts. Signal recorded under conditions identical to those in (**B**) except the cables.

DOI: https://doi.org/10.7554/eLife.30552.006

entering the input range of a standard EEP amplifier (green area in *Figure 1*), contributing to an extended period of data loss. To address this problem, we developed an electrical shield for the TMS coil that substantially attenuates the coil-emitted electric field (*Figure 3B*). One important consideration in shield construction is that the amount of eddy current in the shield should remain low; otherwise, strong vibration or even magnetic attenuation may occur. Therefore, we applied a layer of weakly conducting material in shield construction and the resulted shield possesses an electrical resistance of 10 kΩ (see Materials and methods) that does not result in vibration and magnetic attenuation.

To verify the performance of our shield, we first used a magnetic pickup probe to confirm that at 10 kΩ, the shield does not attenuate the magnetic output of our TMS coil. As *Figure 3C* illustrates, induction voltage waveforms, with and without the shield, overlap perfectly, confirming the absence of any noticeable magnetic attenuation. Subsequently, using a high-impedance buffer, we measured the voltage between an EEP microelectrode and a ground electrode, both electrode tips immersed in a saline bath, under mspTMS at 100% maximum stimulator output (MSO) delivered with or without the shield. We expected that the shield would remove, to a large extent, voltage signal that is due to electric field coupling between the TMS coil and the EEP recording assembly. *Figure 3D* illustrates the results from these measurements. Without the shield, the captured waveform was drastically different from the induction waveforms resulted by mspTMS (as seen in *Figure 3C*), and it ended with a strong voltage bias (polarization). With the shield in place, the captured waveform appeared consistent with the induction waveforms and the voltage bias was no longer visible. These findings confirm that by interrupting electric field coupling, the shield is effective in preventing polarization and the decay artifact that follows.

## Attenuation of vibration artifacts

Upon elimination of artifacts resulted from induction and electric field coupling, vibration artifacts, which are normally masked by the other artifacts, become visible (*Figure 4B*). Vibration can be generated by magnetic force, as well as by sound pressure perturbation. For the magnetically mediated vibration, an avoidance of ferromagnetic materials and large conductive surfaces in the close vicinity of the coil is adequate. However, the elimination of vibration artifacts driven by sound pressure is

not straightforward. When a TMS pulse is triggered, a loud click sound is produced by coil wires due to the attractive forces between them. This sound is problematic as it generates micro-vibration in the amplifier input cables. The extremely weak signal (μV-level) these cables carry can be easily perturbed by micro-vibrations through the triboelectric effect (*Fowler, 1976*). Since the generation of such click sound is inevitable, we attenuated the vibration artifacts by using a special type of low-noise miniature coaxial cable with a semiconducting layer added between its dielectric and braided shield (*Figure 4A*). The addition of this semiconducting layer provides a path that drains triboelectric charges, rendering the cables insensitive to vibration (*Figure 4C*).

Despite the impressive performance of the low-noise miniature coaxial cable in attenuating vibration artifacts, its length in an EEP recording assembly should be limited as the cable's capacitance (100 pF/m), together with the amplifier input capacitance and electrode impedance, acts as a voltage divider that attenuates EEP signal. In our case, we kept the length of our cables at 16 cm to keep the signal attenuation less than 20% at 1 kHz.

## Minimization and determination of inadvertent charge injection

TMS-driven inadvertent charge injection is another major issue, which has been overlooked by most prior reports using EEP under TMS (*Moliadze et al., 2003*, *2005*; *Pasley et al., 2009*). By inserting electrodes into the brain and connecting them to the measurement electronics, multiple loops of electric circuit are formed (*Figure 5*). When being subjected to alternating electric and magnetic field, voltages can be readily developed along these loops that drive unwanted current injection into the brain through microelectrodes. If the amount and the temporal structure of the injected current are comparable to the threshold parameters reported in intracortical microstimulation (ICMS) literature (bipolar charge transfer totaling from 150 to 800 pC, current waveform approximately similar to that of TMS; see *Asanuma and Rosén, 1973* and *Butovas and Schwarz, 2003*), such current will excite neuronal elements around the microelectrode tips and therefore severely confound the measurement of TMS effects. Therefore, it is crucial that the development of voltages along these loops be minimized. Since a large portion of the TMS-emitted electric field had already been filtered away

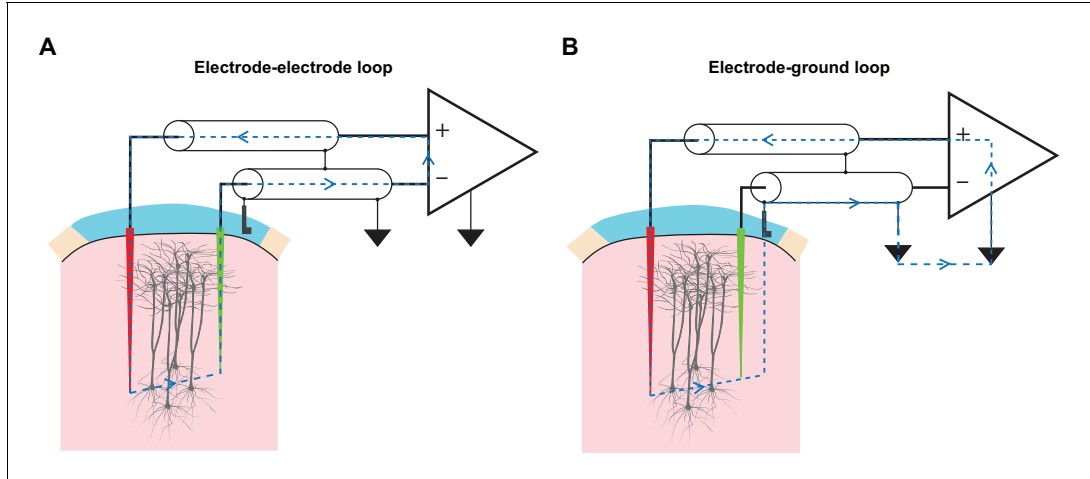

**Figure 5.** TMS drives inadvertent charge injection in multiple loops formed by an EEP recording assembly. (**A**) A schematic illustration of the induction loop formed between the recording (red) and the reference (green) microelectrode under TMS. In the case that TMS-induced voltage is high enough, a substantial amount of electrical current (blue dashed lines) can flow within the loop and subsequently, this may inadvertently stimulate neuronal elements around the microelectrode tips. (**B**) A schematic illustration of the induction loop formed between a microelectrode and the ground electrode under TMS.

DOI: https://doi.org/10.7554/eLife.30552.007

The following figure supplements are available for figure 5:

**Figure supplement 1.** The three-pronged electrode set design.
DOI: https://doi.org/10.7554/eLife.30552.008

**Figure supplement 2.** Circuit representations of the two induction loops shown in *Figure 5*.
DOI: https://doi.org/10.7554/eLife.30552.009

by the coil shield, precautions were taken for magnetic induction. These included a compact arrangement of microelectrodes as well as cable twisting (*Figure 5—figure supplement 1*), both minimize the area of induction loops exposed to TMS. More importantly, we incorporated a low-gain monitoring channel (LGM as seen in *Figure 2*) in our amplifier design that allowed us to conveniently determine the amount of inadvertent current flow, without any additional measurement devices, under each experimental setup. The conversion from voltage, which is measured by the LGM, to current is made possible since amplifier's input capacitance and its voltage fluctuation are known, and the amount of current flow through the input capacitance is equal to the amount of current flow in the circuit. A detailed description of this conversion is presented in Materials and methods and *Figure 5—figure supplement 2*.

## In vivo method evaluation under various types of TMS stimuli

In six male Sprague-Dawley rats, we evaluated and optimized our method. *Figure 6A* offers an overview of our recording setup and the subsequent figures illustrate the performance of the method in vivo under a single monophasic (*Figure 6B*) and biphasic (*Figure 6C*) TMS pulse, as well as a triplet of 50 Hz biphasic pulses (*Figure 6D*), which is the fundamental building block of theta burst stimulation (*Huang et al., 2005*). The stimuli delivered here can be considered as the 'worst-case scenarios' as the stimulator-coil combinations used yield magnetic outputs (peak strength up to four tesla) that are one of the highest among commercially available TMS systems (see Materials and methods). Nonetheless, the EEP signal recovered between 0.8 and 1 ms after the onset of each TMS pulse and was free from artifacts. Furthermore, the amount of inadvertent charge injection under each condition was far below (by a factor of 200 or more; *Figure 6—figure supplement 1*) the modulation or activation thresholds reported in ICMS literature (*Asanuma and Rosén, 1973*; *Butovas and Schwarz, 2003*), confirming the validity of our measurements.

## mspTMS evokes in the layer V of forelimb M1 a multiphasic rhythm of neuronal activities

With the newly developed method, we sought to address the question: what are the neuronal correlates of TMS cortically evoked muscle activation? In another group of seven male Sprague-Dawley rats (anesthetized by ketamine-xylazine), we recorded in vivo mspTMS-evoked neuronal activities in the layer V (*Figure 7—figure supplement 1*) of the caudal forelimb area (CFA), rodent's equivalent to the forelimb area of primate M1 (*Rouiller et al., 1993*). With the coil center positioned over the left CFA and the induced current pointing from the medial to the lateral part of the brain (ML stimulation; *Figure 6A* inset), mspTMS evoked unilateral movement of the right forelimb. Simultaneous intramuscular electromyogram (EMG) recordings of the left and right biceps brachii muscle revealed motor unit action potentials (MUAPs) unilaterally in right biceps brachii (contralateral to the stimulated hemisphere; *Figure 7B* and the insets of *Figure 7C–F*). The onset latency of the MUAPs was around 11 ms, similar to that found in our single-pulse ICMS experiment (*Figure 7—figure supplement 2*) and in the rodent single-pulse ICMS literature (*Liang et al., 1993*; *Deffeyes et al., 2015*), confirming the cortical origin of TMS-evoked muscle activation.

At the neuronal level, in layer V of the CFA, mspTMS evoked a rhythm of neuronal activities alternating between excitation and inhibition that lasted until approximately 300 ms post-stimulation. *Figure 7A* illustrates the multiunit spike raster and its corresponding peristimulus time histogram (PSTH) of multiunit firing rate (FR) from one animal. *Figure 7C–F* show the evoked normalized FR (instantaneous FR subtracted by baseline average FR; see Materials and methods) with increasing stimulation intensity (0%, 95%, 100%, and 120% motor threshold, MT) averaged across all animals. Significance thresholds were drawn based on the 2.5 and 97.5 percentile of the empirical distribution of normalized FR during baseline (500 ms pre-TMS; see Materials and methods for details) to control Type I error rate ($p<0.05$). We categorized the evoked significant excitatory and inhibitory events into three phases: intermediate excitation (a period of increased FR that peaks around 20 ms), inhibition (a long-lasting pause in FR after the intermediate excitation), and rebound excitation (a period of increased FR following the inhibition). To investigate the effects of stimulation intensity on the normalized FR of each phase, we constructed hierarchical linear mixed-effects models. Stimulation intensity positively modulated the normalized FR for the intermediate excitation phase ($\beta = 2.75 \pm 0.24$, $F(1)=127.23$, $p<0.001$) and the rebound excitation phase ($\beta = 1.18 \pm 0.19$, $F(1)$

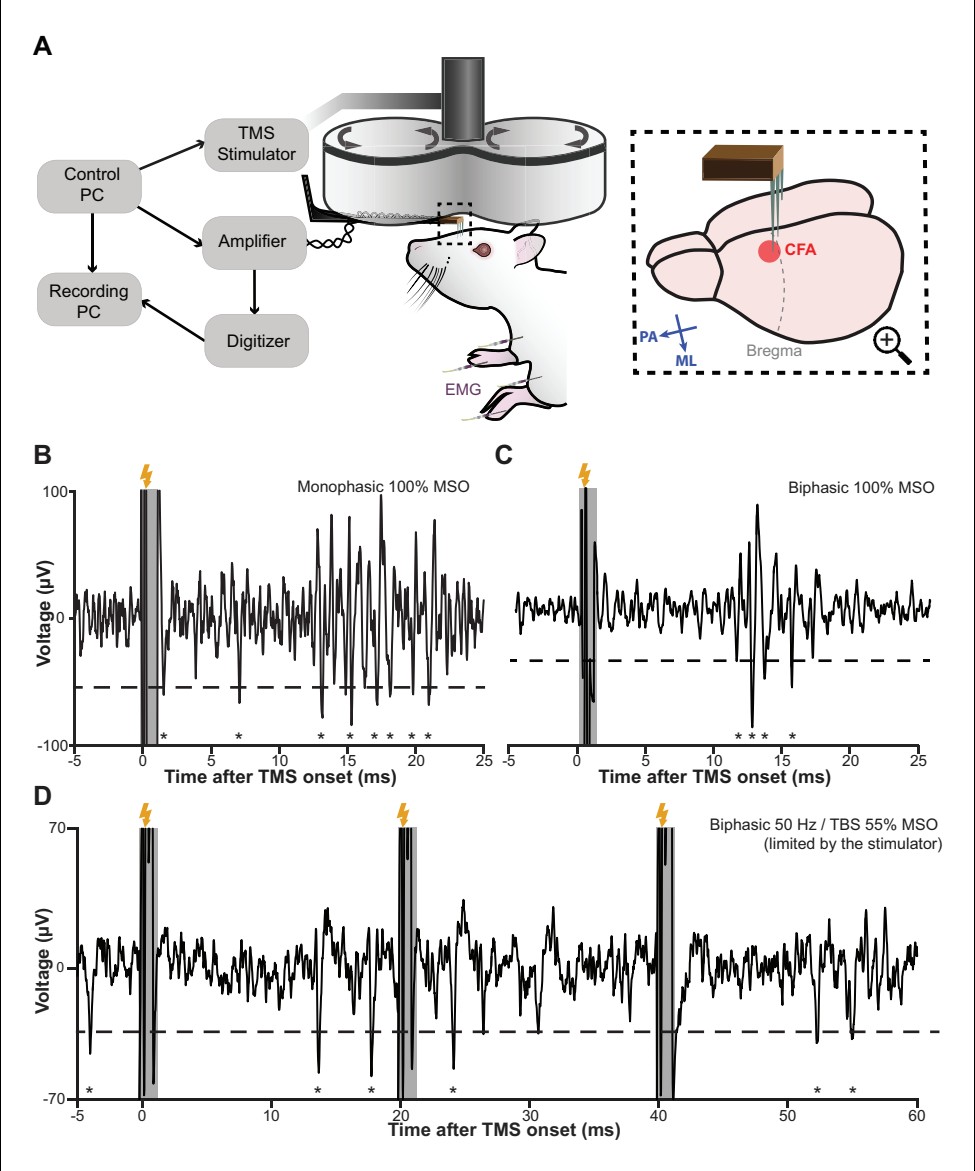

**Figure 6.** TMS-EEP recording setup and rapid signal recovery under the worst-case TMS stimuli. (**A**) A schematic illustration of our recording setup. Thick arrows, direction of coil current; inset blue arrows, direction of induced current in the brain; ML, medial-lateral; PA, posterior-anterior; CFA, caudal forelimb area (rodent's equivalent to forelimb M1 in primates); EMG, intramuscular electromyography. (**B–D**) A sample trace of in vivo recordings under the worst-possible (see Materials and methods) monophasic, biphasic, and theta-burst (first three pulses) stimulus, respectively. The short transient (−0.2 to +0.8 ms) during which the amplifier is protected from the induction artifact is indicated in gray. Lightning symbol, TMS onset (at 0 ms); dashed line, spike detection threshold (see Materials and methods); asterisks, extracellular spikes.

DOI: https://doi.org/10.7554/eLife.30552.010

The following figure supplement is available for figure 6:

**Figure supplement 1.** In vivo measurements of inadvertent charge injection.

DOI: https://doi.org/10.7554/eLife.30552.011

=38.65, p<0.001), while negatively modulated the normalized FR for the inhibition phase (β = −0.23 ± 0.10, F(1)=5.28, p=0.02). It is important to note here that despite the faithful EMG response in the contralateral biceps brachii muscle, the neuronal firing rate in the short-latency window (1–6 ms after TMS onset) was low. This finding was rather surprising and will be further explored in the following section.

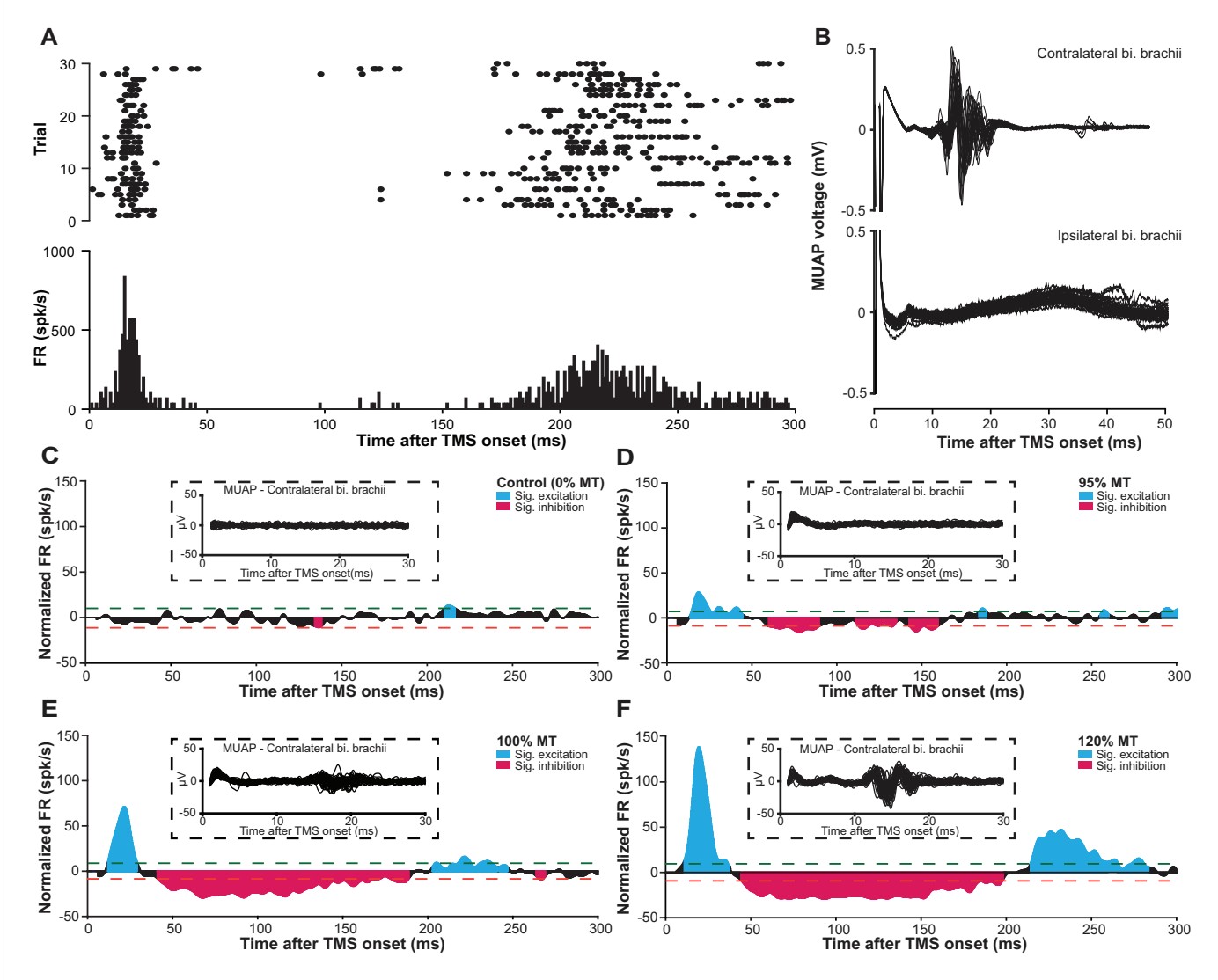

**Figure 7.** mspTMS evoked multiphasic response alternating between excitation and inhibition. (**A**) Raster plot (top) and PSTH (bottom; binsize 1 ms) of multiunit spike activity evoked by mspTMS (stimulus orientation ML; intensity 120% MT; onset at 0 ms) recorded in layer V of the CFA from one animal. (**B**) Traces of evoked MUAPs (corresponding to trials in A) obtained by intramuscular EMG in the biceps (bi.) brachii muscle contralateral and ipsilateral to the stimulated CFA. (**C–F**) Population average (N = 7) of normalized multiunit FR in the layer V of CFA evoked by ML-oriented mspTMS of increasing intensity. The PSTHs were smoothed by a Gaussian kernel for visualization. Inset, example traces of evoked MUAP in the contralateral bi. brachii from one animal. Dashed lines, significance thresholds determined by the 2.5 or 97.5 percentile of the empirical distribution of baseline normalized FR (see Materials and methods for details).

DOI: https://doi.org/10.7554/eLife.30552.012

The following figure supplements are available for figure 7:

**Figure supplement 1.** Histological confirmation of electrode placement.
DOI: https://doi.org/10.7554/eLife.30552.013

**Figure supplement 2.** MUAP evoked by single-pulse ICMS.
DOI: https://doi.org/10.7554/eLife.30552.014

**Figure supplement 3.** Layer V neuronal response evoked by PA-oriented mspTMS at different intensities.
DOI: https://doi.org/10.7554/eLife.30552.015

**Figure supplement 4.** mspTMS evoked a multiphasic pattern of neuronal response in layer II/III.
DOI: https://doi.org/10.7554/eLife.30552.016

## mspTMS evoked short-latency (1–6 ms) neuronal responses differ with stimulus orientations

Since we did not observe any significant modulation of neuronal FR in the short-latency window (1–6 ms) after mspTMS despite faithful muscle activations in the contralateral forelimb, in another set of experiments (N = 4), we explored the possibility that neuronal response in this very early time window is dependent on the direction of mspTMS-induced current. In this set of experiments, we switched the TMS coil orientation so that the induced current flows from the posterior to the anterior part of the brain (PA stimulation; *Figure 6A* inset). We could replicate most findings found in the previous set of experiments as the multiphasic response evoked by PA stimulation is qualitatively similar to that evoked by ML stimulation (*Figure 7—figure supplement 3*). However, in the short-latency window after TMS onset, the neuronal responses observed in ML and PA stimulation are drastically different. As the two examples in *Figure 8A* demonstrate, at 120% MT intensity, ML stimulation evoked scarcely any spike, whereas PA stimulation evoked robust neuronal firing generating a distinct temporal pattern with peaks at 1.2–1.6 ms and at 3.2–4.2 ms. To quantify the short-latency responses in the population, we constructed PSTHs of normalized FR across all animals under ML (*Figure 8B*) and PA (*Figure 8C*) stimulation. Significance thresholds were drawn using the 2.5 and 97.5 percentile of normalized FR distribution during baseline. ML stimulation evoked no significant excitation with the exception of the low albeit significant FR at 3.5–4 ms under stimulation intensity of 120% MT. On the contrary, under PA stimulation, multiple significant excitatory events were observed. At subthreshold level, significant excitatory events appeared at 2.5–3.5 ms and at 4–4.5 ms. As the stimulation intensity increased, FR was developed at particular time windows: 1–1.5 ms and 2.5–4.5 ms, reminiscent of the indirect wave (I-wave) phenomena observed in the corticospinal descending volleys in human and animal studies.

## Discussion

Our understanding of the neuronal mechanism of TMS has been largely based on indirect evidence obtained at the level of cortical output reflected in spinal cord or muscle activities. Direct investigation of the dynamics of neuronal activities evoked by TMS was hindered by technical obstacles imposed by the strong electromagnetic pulse produced by TMS. We engineered a widely applicable experimental method for the in vivo study of TMS-evoked brain activities at the level of neurons using EEP. It allows the monitoring of neuronal activities as early as 0.8–1 ms after the strong electromagnetic perturbation of various TMS stimuli ranging from single pulse to the high-frequency theta burst stimulation. Our method encompasses solutions to all major challenges in concurrent TMS-EEP recording, including magnetic induction, electric field coupling, vibrations, and inadvertent charge injection. Despite the multidimensional approach of our method, it was developed with generalizability, simplicity, flexibility, and scalability in mind. It is compatible with, but not limited to, rodents, an animal model that is widely used for studying basic neurophysiology and offers a wide range of investigative tools. It does not require active compensation based on magnetic field sensing (*Logothetis et al., 2001*; *Mueller et al., 2014*) or custom-made coils (*Tischler et al., 2011*; *Mueller et al., 2014*) for artifact reduction. It can accommodate electrodes directly under a conventional TMS coil, a feature making it suitable even for awake animals with chronically implanted electrodes. In addition, the method can be scaled up for large-scale high-density EEP recording with silicon-based microelectrode arrays (*Buzsáki, 2004*) as well as be accompanied by optogenetic tools for the in vivo control of neuronal circuits (*Scanziani and Häusser, 2009*).

The amount of magnetic, electric, and vibrational interference TMS imposed on EEP depends on multiple factors. Some of the most critical factors include the waveform and magnitude of the pulsed magnetic and electric field emitted from a TMS coil, the size of circuit loops formed by an EEP recording assembly, and the coil position relative to these loops. Changes in these parameters will result in changes in the severity of different types of interference. For example, keeping the coil the same, by replacing a standard monophasic with a standard biphasic stimulator, coil-emitted fields will generate a longer period of magnetic and electric field interference due to the longer pulse waveform. However, the severity of interference might be lower if the coil and biphasic stimulator combination does not produce magnetic and electric outputs that are as high as those in the monophasic case. Similarly, miniaturization of TMS coils for small animals can also lead to a reduction in interference because of the reduced electromagnetic output of such devices. Furthermore, the

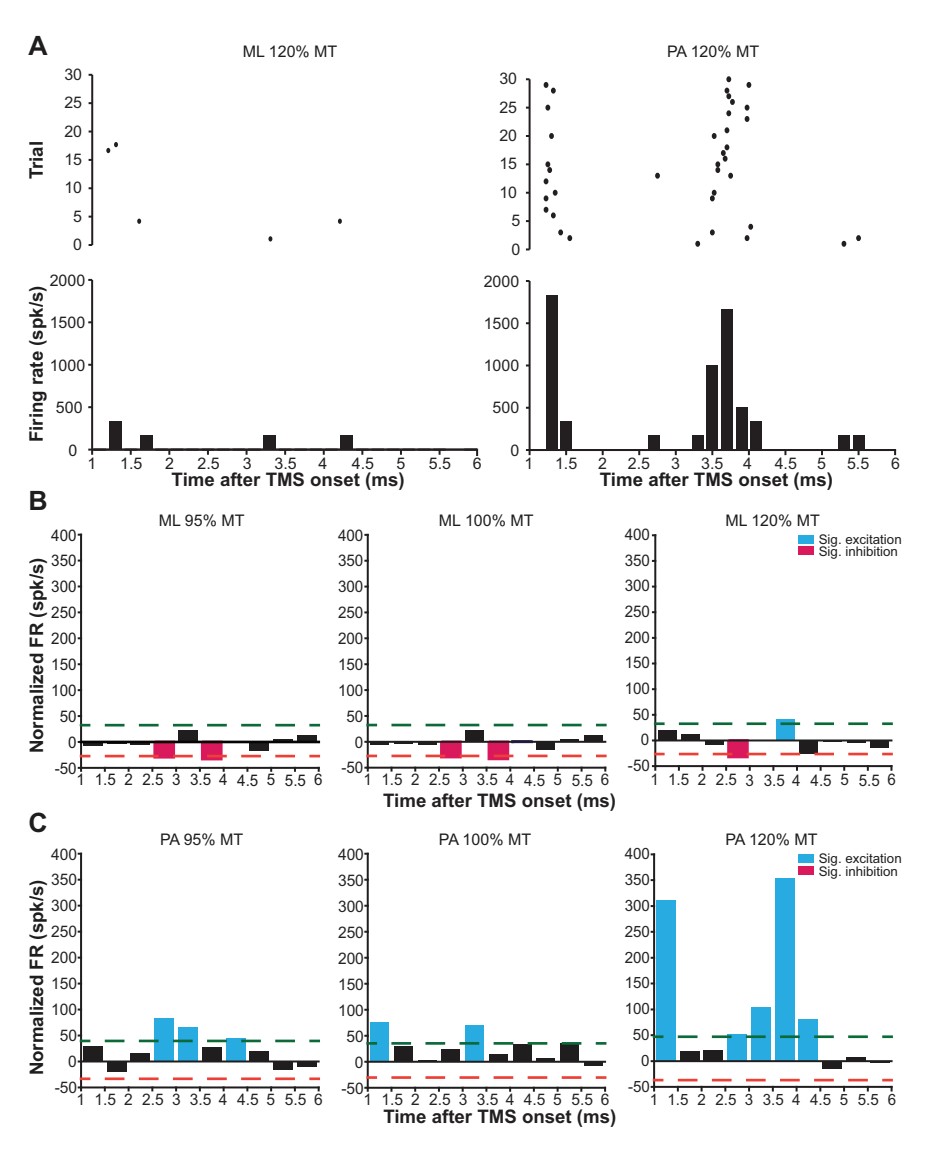

**Figure 8.** mspTMS-evoked short-latency neuronal responses differ with stimulus orientations. (**A**) Examples of short-latency (1–6 ms) multiunit response in layer V of the CFA to suprathreshold mspTMS (120% MT) oriented in ML and PA direction. The suprathreshold ML stimulus evoked virtually no response in this time window, whereas the PA stimulus evoked strong periodic firing in the neuronal population. Note, all orientations discussed here refer to the orientation of induced current in the brain. (**B–C**) Average of short-latency normalized multiunit FR (binsize 0.5 ms) across all animals tested with the ML- (**B**) and PA- (**C**) oriented mspTMS at increasing intensities. Dashed lines, significance thresholds determined by the 2.5 or 97.5 percentile of the empirical distribution of baseline normalized multiunit FR (see Materials and methods for details).

DOI: https://doi.org/10.7554/eLife.30552.017

integration of recording, reference, and ground electrode in one microfabricated electrode array can also reduce the severity of interference as such configuration significantly decreases the area of circuit loops exposed to TMS.

One common criticism of TMS investigations in rodents is that the TMS coil is large compared to the size of a rodent brain. While we fully acknowledge this concern, we argue that it is not a problem of critical importance at this stage. With careful coil positioning, it is possible to achieve certain level of spatial selectivity as evident in the results of our study as well as those of several other reports (*Nielsen et al., 2007*; *Rotenberg et al., 2010*; *Muller et al., 2014*). In addition, plasticity, assessed

by motor output (*Muller et al., 2014*), learning performance (*Mix et al., 2010*), sensory-evoked neural activities (*Thimm and Funke, 2015*; *Murphy et al., 2016*), or protein expressions (*Trippe et al., 2009*; *Benali et al., 2011*), can also be successfully induced in rodents using human TMS coils, making rodents a suitable experimental model for investigating the basic neuronal mechanisms underlying stimulation-induced plasticity. Furthermore, TMS can be used as a tool to deliver a strong transient stimulus to perturb neuronal populations of the neocortex (*Walsh and Cowey, 2000*). Being able to capture the neuronal response to such perturbation at spike resolution will undoubtedly open up another avenue to study the connectivity and the functional properties of neuronal networks. Nonetheless, the development of smaller and more compact coils specifically designed for small animals would be beneficial for their improved spatial resolution and smaller electromagnetic interference as the maximum magnetic output of these coils is much smaller (at mT level; *Makowiecki et al., 2014*; *Tang et al., 2016*) than the 4T output tested in our development.

To validate our method, we successfully translated mspTMS to rodents and unveiled the evoked neuronal activities underlying this classical TMS stimulus which has been widely used in humans since its introduction in 1985 (*Barker et al., 1985*). At the behavioral level, mspTMS delivered in either ML or PA direction over left CFA evoked unilateral forelimb movement in the contralateral side. The similarity between the onset latency of mspTMS- and single-pulse ICMS-evoked MUAPs suggests the cortical origin of the TMS-evoked muscle activations. At the neuronal level, despite the similar evoked motor outputs, mspTMS delivered in the ML and PA orientation evoked different CFA layer V neuronal activities in the short-latency window (1–6 ms) after TMS onset. While threshold or suprathreshold ML oriented stimuli evoked virtually no response in this time window, PA oriented stimuli evoked population spiking activities that occurred preferably around 1–1.5 ms and at 3–3.5 ms (*Figure 8*). Such discrepancy in neuronal response suggests that TMS-induced current of different orientations activates different microcircuits in the rodent forelimb M1: ML-oriented stimuli directly activated pyramidal cells of the descending motor pathways while PA-oriented stimuli evoked transsynaptic high-frequency spiking activities in M1 (*Figure 9A*). It is important to note here that neuronal activity within 1 ms after TMS onset is not visible. Therefore, any antidromic spike evoked by direct axonal activations could not be recorded.

It might be argued that the observed discrepancy in short-latency response is a result of bias in neuronal sampling. We believe this is rather unlikely, as short-latency spikes evoked by ML stimulation were absent across multiple recording sites within CFA (0 out of 7 sites) while the significant high-frequency spiking pattern was observed readily within CFA under PA stimulation (3 out of 4 sites). Additionally, in PA trials, the observed high-frequency spiking disappeared when we turned the stimulus orientation to ML. While we cannot rule out the possibility that mspTMS evoked early spike responses in areas other than the ones we monitored, our data supports the notion that in the layer V of CFA — the output layer of the rodent forelimb M1 — selectivity in stimulus orientation exists. Another confounding factor that might explain the discrepancy is the intensity difference of induced electric fields in the brain under ML and PA stimulation. Since the rodent skull is not spherical, with a given coil output, induced electric field in the ML direction (along the short axis of the skull) should be lower in intensity than that in the PA direction (along the long axis of the skull), raising the possibility that the observed high-frequency spiking pattern under PA stimulation is a result of high intensity of the induced electric field. However, motor thresholds under ML stimulation, in which induced electric field intensity is lower, were significantly lower than their PA counterparts (median$_{ML}$ = 61% MSO; median$_{PA}$ = 74% MSO; Wilcoxon rank-sum test, p=0.03). This is a strong indication that factors other than induced electric field intensity play a critical role in stimulus orientation selectivity. Therefore, we conclude that the observed response difference between ML and PA stimulation is unlikely to be caused solely by the difference in the intensity of induced electric fields.

TMS works on human (*Kaneko et al., 1996*; *Di Lazzaro et al., 2001*) and non-human primates (*Amassian et al., 1990*; *Amassian and Stewart, 2003*) also reported similar stimulus orientation selectivity but in the context of evoked motor outputs. However, we stress that the similarities between our results and those of humans and non-human primates rest solely at the level of a shared common principle: TMS-evoked direct activation is a product of the interaction between TMS-induced electric field and the anatomical and physiological properties of the neurons within. Despite different levels of complexity between primate and rodent brains, certain neuronal structures are preferably stimulated in one stimulus orientation rather than the others. But whether such similarity is based on shared anatomical and physiological properties warrants further investigation.

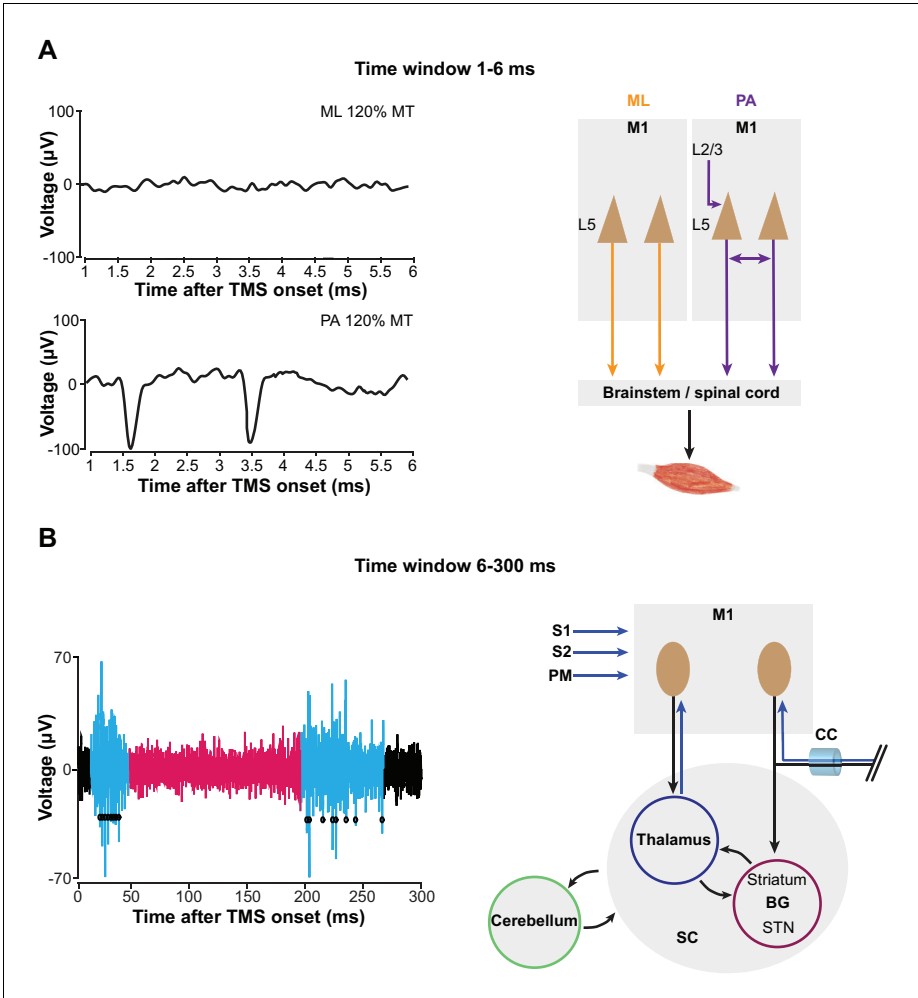

**Figure 9.** mspTMS activates different neuronal circuits depending on stimulus orientation or the time-window of investigation. (A) In the short-latency time window (1–6 ms after onset), ML- and PA-oriented mspTMS evoked different patterns of neuronal activities in layer V of CFA (left panel). ML stimuli activated the descending PT pathways, while PA stimuli triggered an oscillatory spiking event that reflects the local connectivity within M1 (right panel). (B) In the long-latency time window (6–300 ms after onset), mspTMS evoked a multiphasic response alternating between excitation and inhibition (left panel shows a raw spike trace evoked by a suprathreshold stimulus; blue and red code for phase of significant excitation and inhibition, respectively, adopted from *Figure 7*). This multiphasic pattern is generated through multiple possible long-range circuits activated by mspTMS (right panel). Abbreviations: BG, basal ganglia; CC, corpus callosum; M1, primary motor cortex; PM, premotor cortex; S1/S2, somatosensory cortices; SC, subcortical structures; STN, subthalamic nucleus.
DOI: https://doi.org/10.7554/eLife.30552.018

Furthermore, the primate cortex is gyrencephalic while the rodent cortex is lissencephalic. As we could reliably stimulate a lissencephalic M1 and evoke muscle activation on the contralateral fore-limb at the correct cortically evoked latency, the locus of direct TMS activation is most likely not dependent on the magnitude of induced electric field component normal to the cortical columns (*Fox et al., 2004*; *Bungert et al., 2017*).

The evoked short-latency response in the PA orientation was characterized by population spikes at a very high frequency similar to that of the I-waves recorded in the corticospinal tracts of humans and animals in response to a transient shock delivered to the M1 by either transcranial electrical stimulation (*Patton and Amassian, 1954*; *Kernell and Chien-Ping, 1967*) or TMS (*Kaneko et al., 1996*; *Nakamura et al., 1996*; *Di Lazzaro et al., 2012*). What are the principles of anatomical and functional organization in M1 that drive such high-frequency neuronal response? We recorded in

layer V of the motor cortex (*Figure 7—figure supplement 1*) where two types of excitatory projection neurons exist: the corticospinal tract (PT) neurons that project to midbrain, brainstem, and spinal cord, and the intratelencephalic (IT) neurons that project ipsi- or bilaterally within the cortex and striatum (*Harris and Shepherd, 2015*). It has been shown recently that PT neurons exhibit reciprocal connectivity characterized by short-term facilitation and that synaptic transmission time for a pair of reciprocally connected PT neurons is 1.6 ± 0.5 ms (*Morishima and Kawaguchi, 2006*; *Morishima et al., 2011*). Therefore, it is plausible that the network formed by the interconnected PT neurons in layer V provides the physiological foundation for the high-frequency neuronal discharge and that a mspTMS pulse oriented in PA direction preferably delivers an input into this network that triggers the observed high-frequency spiking response (*Figure 9A*). Furthermore, the interconnected PT network may also offer a neuronal explanation for the short-interval intracortical facilitation (SICF) described in the human literature (*Tokimura et al., 1996*; *Ziemann et al., 1998*).

As we extend the window of investigation to 6–300 ms after TMS onset, a multiphasic response appears among the recorded CFA layer V neurons. The response is characterized by its excitation-inhibition-excitation pattern and is not qualitatively different between PA and ML stimulations (*Figure 7*; *Figure 7—figure supplement 3*). The strong excitation that peaks around 20 ms, given its latency, duration, presence in both layer V and II/III (*Figure 7—figure supplement 4*), and its apparent lack of motor output (*Figure 7B*), reflects a high excitability state of the motor cortex. We hypothesize that this excitation is generated through the cortico-basal ganglia-thalamo-cortical loops (*Figure 9B*). Evidence suggests that cortex projects monosynaptically to basal ganglia (BG) structures such as striatum and subthalamic nucleus (STN) (*Kita and Kita, 2012*), while the projection from striatum and STN back to cortex is polysynaptic (*Shepherd, 2013*). Deep brain stimulation (DBS) of the STN produces cortically evoked EEG potentials with a peak latency of 22.2 ± 1.2 ms, and TMS delivered at this latency after DBS showed facilitation of its cortically evoked motor outputs (*Kuriakose et al., 2010*). It is likely that TMS activates IT and PT neurons that project to BG monosynaptically, and the response is then transmitted back to the cortex as the intermediate excitation observed here. But other cortico-cortical or cortico-subcortical loops could be involved as well. The neuronal mechanism of TMS protocols such as intracortical facilitation (*Ziemann et al., 1996*) and theta burst stimulation (interpulse interval of 20 ms within each burst) (*Benali et al., 2011*; *Suppa et al., 2016*) remain unknown; however, it is conceivable that these protocols exploit this particular phase of excitation for their physiological effects. The long-lasting inhibition phase that follows the intermediate excitation is well-known, and evidence supports the notion that it is mediated by GABA$_B$ (*Butovas et al., 2006*; *Murphy et al., 2016*) and underlies the long-interval intracortical inhibition as well as the cortical silent period in human TMS (*Valls-Solé et al., 1992*; *McDonnell et al., 2006*). However, the local or long-distance circuit mediating this phase of inhibition remains unknown. The rebound excitation phase, occurring after the inhibition, represents a period of excitation most likely resulting from the termination of GABA$_B$ inhibition, and corresponds to the late cortical disinhibition, which is being harnessed for augmenting plasticity induction in human TMS (*Cash et al., 2016*). Similarly, the circuit mechanism behind this phase of rebound excitation remains to be elucidated as well.

Would the same neuronal activity pattern be observed if a rodent-sized TMS coil is used to stimulate the forelimb M1? We believe that this is the case since we carefully calibrated coil position and stimulation strength according to MEP. Furthermore, the long-lasting inhibition and the rebound excitation are well-documented phenomena in ICMS (*Butovas and Schwarz, 2003*), which is a much more localized stimulation method than TMS. Additionally, as discussed above, data from human TMS is largely congruent with the pattern of neuronal activity reported here. However, we cannot rule out the possibility that the coil we used in this study directly activated structures outside of the forelimb M1. Nonetheless, the role of stimulus spatial resolution in modulating neuronal networks is a highly interesting topic for future research.

By combining the tool presented here with optogenetic, transgenic, anatomical, theoretical, and clinical methods, future work could take on two parallel directions concerning either the short- or long-latency evoked response of TMS. For the short-latency response, investigation could focus on discerning the circuit selectivity of different stimulus orientations by pinpointing the locus of direct activation in each case, and on elucidating the principles of anatomical and functional organization of the M1 microcircuitry. Additionally, utilizing TMS as a probe, other cortical areas can be investigated in a similar manner. For the long-latency response, the focus shall be on characterizing long-

range circuits activated by TMS and examining their modulatory contributions in the treatment of various neurological and psychiatric conditions. We are convinced that studying the neuronal dynamics under TMS will undoubtedly advance our understanding of the functional organization of the brain, and drive the development of non-invasive brain stimulation therapies that are more specific, effective, durable and safe than hitherto possible.

## Materials and methods

### Determination of the inadvertent charge injection

To determine the amount of TMS-induced charge injection in the electrode-electrode loop (*Figure 5A*), we used the voltage signal from the low-gain monitoring channel to calculate the current flow via the amplifier's input capacitance $C_{in}$. As shown in *Figure 5—figure supplement 2A*, since the input resistance $R_{in}$ and the input capacitance $C_{in}$ are parallel, voltage drop across $R_{in}$ (therefore, the recorded signal $V_{in}$) is equal to the voltage drop across $C_{in}$. Because the value of $C_{in}$ is known, its current $I_{Cin}$ can be calculated using the equation

$$I_{Cin} = C_{in}\frac{dV_{in}}{dt}$$

Furthermore, since $R_{in}$ is in the order of teraohm, the amount of current it draws can be neglected. Therefore, $I_{Cin}$ is equal to the total amount of induction current present in the loop ($I_{ind}$). It is worth noting here that by adopting this method, the exact model of microelectrodes and its associated component values are not needed for the calculation.

For determining the induced charge injection in the electrode-ground loop, we used a set of input cables in which both the recording and the reference electrode were connected to the amplifier's positive input, and the ground electrode was connected to the amplifier's negative input. Furthermore, the negative input was shorted to the amplifier ground. Under this configuration, $I_{ind}$ reflected the current in the electrode-ground loop (*Figure 5—figure supplement 2B*).

At the end of our validation, we conducted these measurements in vivo, under monophasic and biphasic TMS, at 100% MSO. By integrating $I_{ind}$ over time, the amount of charge transfer was determined. The results (*Figure 6—figure supplement 1*) were then compared with the charge injection values reported in the ICMS literature.

### Electrical shield

To construct the electrical shield (*Figure 3B*), we first made a polyoxymethylene (POM) enclosure (1 mm thick at the bottom face) according to the shape of our TMS coil. An even layer of conductive coating (GRAPHIT 33, Kontakt Chemie, Iffezheim, Germany) was painted on the inner side of the enclosure until the desired electrical resistance (10 kΩ measured along the long axes of the shield body and cover) was reached. A layer of non-conductive transparent coating was then applied to protect the conductive layer. As the body and the top covers of the enclosure are separate, protection coating was not applied along the contacting edges between the shield body and its top covers to allow good electrical contact. In addition, an electrical cable was connected directly to the conductive layer to provide a path for grounding.

### Experimental model

All experimental procedures involving animals were approved by the Tübingen Regional Council (license number: N1/16) and performed in accordance with the Animal Welfare Act of Germany. Seventeen male Sprague-Dawley rats (Charles River Laboratories, Sulzfeld, Germany; RRID:RGD_737891) 11–15 weeks of age were used (six for method evaluation and optimization; seven for the ML experiments; four for the PA experiments). The animals were housed in environment-enriched transparent plastic cages under inverted 12 hr light/dark cycle with free access to water and food. Upon arrival, the animals were handled 10 min per day for 5 consecutive days for stress reduction.

### Surgery

Animals were first sedated through a brief exposure to isoflurane (3% at 0.8 L/min). Upon sedation, a cocktail of ketamine (70 mg/kg) and xylazine (1 mg/kg) was injected intraperitoneally (i.p.) and

ophthalmic ointment was applied to eyes. A 27-gauge catheter was implanted in the lower right quadrant of the abdomen to provide i.p. access throughout the experiment. Additional doses of ketamine (30 mg/kg) were administered through the catheter to maintain a constant level of anesthesia, which was assessed by breathing rate, vibrissa whisking, and the toe-pinch reflex. During the incision phase of the surgery, xylocaine gel (2%) was applied to the incision site. In addition, body temperature of the animals was maintained at 37°C using a feedback-controlled heating pad throughout the experiment.

Animals were restrained in a stereotaxic frame with non-conductive ear bars. A $5 \times 3$ mm craniotomy was made over the left sensorimotor cortex. The resulted trepanation extended from $-1$ mm to $+4$ mm to bregma and from 1 mm to 4 mm lateral to the midline. Dura matter was carefully resected and the cranial window was covered with Ringer's solution.

## Intracortical microstimulation (ICMS)

ICMS was used to map the spatial extent of the primary forelimb motor area (caudal forelimb area, CFA). A platinum-tungsten microelectrode (1 MΩ at 1 k Hz) was used for ICMS at depths around 1400 μm (from the cortical surface), corresponding to layer V in rat neocortex, with a train of 13 biphasic square pulses (200 μs per phase) delivered at 333 Hz. A stimulation site was considered non-responsive if it was not possible to elicit any visible movement with current intensity up to 100 μA. In one animal, we also used single-pulse ICMS (one biphasic square pulse, 300 μs per phase, 300 μA) to study the onset latency of MUAP of the biceps brachii in response to ICMS (*Figure 7—figure supplement 2*).

## Electromyogram (EMG)

28-gauge monopolar EMG electrodes (Ambu A/S, Ballerup, Denmark) were implanted in both left and right biceps brachii muscle for recording, and in the finger pads bilaterally for reference. The electrodes were connected to a high-impedance amplifier through shielded cables. The signal was low-pass (cutoff frequency 5 kHz) filtered online and amplified 2000 times before digital conversion. During analysis, the signal was bandpass filtered (100–1000 Hz) using digital Butterworth filters implemented anti-causally in MATLAB.

## Transcranial magnetic stimulation (TMS)

TMS was delivered through a Magstim D25 figure-of-eight coil (single circle radius 25 mm; Magstim Ltd., Carmarthenshire, UK) powered by either a Magstim 200[2] stimulator for monophasic single-pulse stimulation (mspTMS) or a Magstim Super Rapid Plus stimulator (with the inline inductor Magstim 3467) for biphasic single-pulse and repetitive stimulation. The Magstim 200[2] and D25 combination is considered as the worst-case scenario since the resulting flux transient is as high as 4T (based on data supplied by Magstim), which is two to three times higher than the output seen in combinations with larger coils that are routinely used in human stimulation.

The TMS coil was held by a mechanical arm and positioned over the recording site in medial-lateral orientation, generating a current flowing from the medial to the lateral part of the brain (under monophasic stimulation). In the PA orientation, the induced current flows from the posterior to the anterior part of the brain. The coil, controlled by a three-dimensional microdrive, was lowered as much as possible without touching the electrode assembly. The distance from the coil surface to the head of the animal was normally 8–10 mm (including 1 mm due to the coil shield). TMS was triggered digitally by a controller PC, which also digitally controlled the behavior of our EEP amplifier (*Figure 6A*).

## Extracellular electrophysiology (EEP)

EEP was recorded through a pair (signal-reference) of microelectrodes (ca. 1.5 MΩ impedance at 1 kHz) fabricated in-house from glass-coated platinum-tungsten wires (Thomas RECORDING, Giessen, Germany). A thin silver wire with silver-chloride coating was used as the ground electrode. The three electrodes were arranged in a three-pronged design (*Figure 5—figure supplement 1*) that minimized the induction loop area between them. The assembly was held by a non-conductive non-magnetic L-shape holder that was mounted on a micropositioner (David Kopf Instruments, Tujunga, USA). The recording electrode was lowered, through the cranial window, into CFA as determined by

ICMS. The reference electrode was also lowered into the cortex but outside the boundary of CFA. The ground electrode was positioned to be in contact with unresected subcutaneous tissue by the border of the cranial window. Signals from the electrodes were transmitted through a set of 36-gauge low-noise miniature coaxial cables (Axon' Cable S.A.S., Montmirail, France; *Figure 4A*) to the amplifier. The operating mode of the amplifier was controlled by the controller PC as described in the main text. The signal from the amplifier output was digitized (USB-ME64-System, MultiChannel Systems GmbH, Reutlingen, Germany) at 40 kHz and subsequently visualized and stored on a PC. A schematic illustration of the entire recording setup is shown in *Figure 6A*.

## Histology

Upon completion of an experiment, the recording site was marked by an electrolytic lesion (1 cycle of cathode leading 0.1 Hz biphasic square pulse with 10 µA) generated using a microelectrode powered by a waveform generator (STG1002, MultiChannel Systems, Reutlingen, Germany). Subsequently, the animal was deeply anesthetized with sodium pentobarbital (200 mg/kg) and perfused using phosphate buffer (0.1 M) followed by paraformaldehyde (4%). Afterward, the brain of the animal was processed using standard histological procedures. The recording layer was assessed by investigating lesions in hematoxylin and eosin stained coronal sections (*Figure 7—figure supplement 1*).

## Quantification and statistical analysis

Electrophysiological data was processed in MATLAB 2014b (The Mathworks, Natick, USA; RRID: SCR_001622). Spike detection was based on amplitude threshold that was set to 3.5 or 4 times of the median-based background noise standard deviation estimate in order to minimize the influence of high spike rates or amplitudes in biasing spike detection (*Quiroga et al., 2004*). Spike isolation was performed using principal component analysis of the spike waveforms followed by a Gaussian mixture model with Kalman filters that track waveform drifts over time (*Ecker et al., 2014*). A total of 51 single units were isolated ($L5_{ML}$ = 19; $L5_{PA}$ = 14; $L2/3_{ML}$=18); however, since at the present stage we are only interested in characterizing the response of M1 neuronal population to mspTMS, in the following analysis, we combine spikes from all single units as well as those that cannot be reliably isolated into a multiunit cluster.

Each trial was defined by the time interval spanning from 500 ms pre-TMS to 1000 ms post-TMS. Normalized firing rate (FR) was calculated by subtracting the baseline (500 ms period prior to TMS onset) average FR from the instantaneous FR of each time bin (including baseline bins). This normalization procedure was performed on a trial-by-trial basis. For each animal under each stimulation condition, trains of normalized FR were averaged across trials. Thresholds for significant ($p<0.05$) inhibitory and excitatory events were determined by the 2.5 and 97.5 percentile of the empirical distribution of normalized FR during baseline. To facilitate the detection of significant phasic response, each averaged train of normalized FR was filtered by a Gaussian kernel ($\sigma$ = 2 ms). An event is considered as a significant phasic response if the normalized FR exceeds either threshold for more than 10 ms and a gap up to 10 ms is tolerated to accommodate jittering. The onset and duration information of the detected phasic response was then used to extract FR for each phase in each individual trial.

Statistical analysis was performed in R (*Core Team, 2016*; RRID:SCR_001905). Multiple hierarchical linear mixed-effects models were constructed using the lme4 package (*Bates et al., 2015*) to evaluate the effect of stimulation intensity on the normalized FR for each response phase. Stimulation intensity (normalized to %MT) was used as the fixed effect to model trials of normalized FR of each response phase. The animal's identity was used as the random effect (random intercept) to control for intraclass correlation. We also explored the possibility of trial number being another fixed effect. However, it was dropped in the final models as it did not contribute significantly to model's fit. Statistical significance of the fixed effect in each model was evaluated against the corresponding null model using the Kenward Roger-based F-test (*Halekoh and Højsgaard, 2014*)

## Acknowledgements

This study was supported by funding from the Hertie Institute for Clinical Brain Research and the Werner Reichardt Centre for Integrative Neuroscience, University of Tübingen, Germany, and from the Max Planck Society, Germany. We express our gratitude to Oliver Holder (Max Planck Institute for Biological Cybernetics), Ursula Pascht (Uni. Tübingen), Klaus Vollmer and associates (Uni.

Tübingen) for technical support; Tjeerd Dijkstra (Uni. Tübingen) for consultation on statistics; Klaus Funke (Uni. Bochum) for equipment loan and comments; Nikos Logothetis (Max Planck Institute for Biological Cybernetics) for logistical support and comments; Florian Müller-Dahlhaus (Uni. Tübingen), Dominic Kraus (Uni. Tübingen) and Shuchun Li (Beijing Jiaotong Uni.) for comments. MG was supported by: DFG GZ: KA 1258/15-1, European Commision H2020 CogIMon ICT-644727, HFSP RGP0036/2016 and BW Stiftung KONSENS-NHE NEU/007/1.

## Additional information

### Funding

| Funder | Grant reference number | Author |
|---|---|---|
| Max-Planck-Gesellschaft | | Bingshuo Li |
| Werner Reichardt Centre for Integrative Neuroscience | | Bingshuo Li Cornelius Schwarz Martin A Giese |
| Hertie Institute for Clinical Brain Research | | Cornelius Schwarz Martin A Giese Ulf Ziemann |
| Deutsche Forschungsgemeinschaft | KA 1258/15-1 | Martin A Giese |
| European Commission | H2020 CogIMon ICT-644727 | Martin A Giese |
| Baden-Württemberg Stiftung | KONSENS-NHE NEU/007/1 | Martin A Giese |
| Human Frontier Science Program | RGP0036/2016 | Martin A Giese |

The funders had no role in study design, data collection and interpretation, or the decision to submit the work for publication.

### Author contributions

Bingshuo Li, Conceptualization, Resources, Data curation, Software, Formal analysis, Validation, Investigation, Visualization, Methodology, Writing—original draft, Writing—review and editing; Juha P Virtanen, Axel Oeltermann, Resources, Validation, Methodology, Writing—review and editing; Cornelius Schwarz, Martin A Giese, Resources, Supervision, Funding acquisition, Methodology, Writing—review and editing; Ulf Ziemann, Resources, Supervision, Funding acquisition, Writing—review and editing; Alia Benali, Conceptualization, Resources, Data curation, Formal analysis, Supervision, Validation, Investigation, Visualization, Methodology, Writing—original draft, Project administration, Writing—review and editing

### Author ORCIDs

Bingshuo Li http://orcid.org/0000-0002-9024-8354
Cornelius Schwarz http://orcid.org/0000-0003-4725-473X
Alia Benali http://orcid.org/0000-0001-6047-3713

### Ethics

Animal experimentation: All experimental procedures involving animals were approved by the Tübingen Regional Council (license number: N1/16) and performed in accordance with the German Animal Welfare Act.

### Decision letter and Author response

Decision letter https://doi.org/10.7554/eLife.30552.021
Author response https://doi.org/10.7554/eLife.30552.022

## Additional files

**Supplementary files**
• Transparent reporting form
DOI: https://doi.org/10.7554/eLife.30552.019

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
