## [Decision Letter]

Thank you for submitting your article "Lifting the veil on the dynamics of neuronal activities evoked by transcranial magnetic stimulation" for consideration by *eLife*. Your article has been reviewed by three peer reviewers, and the evaluation has been overseen by a Reviewing Editor and Richard Ivry as the Senior Editor. The following individuals involved in review of your submission have agreed to reveal their identity: Risto J. Ilmoniemi (Reviewer #2); Javier Marquez Ruiz (Reviewer #3).

The reviewers have discussed the reviews with one another and the Reviewing Editor has drafted this decision to help you prepare a revised submission.

Summary

The authors present a novel method for TMS coupled with high fidelity extracellular single neuron recordings in anesthetized rats. This protocol's innovations include electrical TMS coil shielding, specialized amplifier circuitry, and specialized cabling. The presented results provide both a proof of principle for the method, and insight into immediate TMS-modulated single unit activity in the rat motor cortex.

Essential revisions

1) Organization and Presentation:

a) Include, in the Introduction, a clear statement of the problematic system interactions, the manifestation of these interactions in the acquired EEP data and short statements about how each is to be ameliorated. This would include the interactions due to induction, electrical field (capacitive) coupling, vibration, and charge injection.

b) Presently the elucidation of the interactions between the TMS and EEP systems and their corrections come in three different places with increasing levels of detail: The Results, Materials and methods section and the Supplemental Section. Please pare this down to two places with much less appearing in the supplemental section.

c) Clarify how the described methods differ from other methods that address these interactions. For instance, consider the following statement: "Two groups reported TMS-EEP methods for non-human primates (Mueller et al., 2014; Tischler et al., 2011); however, the applicability of these methods is very limited as they were developed solely for primate research, which is not best positioned to answer basic questions on neural circuitry, and they involve complex electronics and custom-designed TMS coils for artifact reduction." For this statement, please describe: (i) the methods that they used, (ii) the time window following the TMS pulse in which these methods are effective (0-1 ms as well?), (iii) how effective the methods are, and (iv) why the methods are not "best positioned to answer basic questions on neural circuitry."

d) The manuscript content together with supplementary information has all the elements required for facilitating the proposed methodological improvements by interested researchers; nevertheless, much of this important information is categorized as "supplementary," giving priority to the observed physiological response to TMS (somehow incomplete data, see comments below). Taken into account that this manuscript has been submitted to "tools and resources" category all the important technical details should be included in the body of the manuscript.

e) There are 5 "figures" in the main part and 10 in the supplement. However, in many figures, there are subfigures (A, B, C.…) and in some of the subfigures, there are multiple panels. The total number of labeled figures or subfigures is about 40. This may be too much. Even the main text contains 19 figures or subfigures. One reason for the large number of figures is that there is material in the manuscript for two manuscripts. For instance, while the abstract emphasizes the development of the engineering and measurement methodology, the discussion and much of the other text emphasize the neuroscientific results. Please consider condensing the text and figures to improve the paper's focus.

2) The section describing the second artifact source which begins at the third paragraph of the Results section lacks clarity. This section can be improved if the authors: (i) state that capacitive coupling between the TMS and EEP systems poses a significant problem which can be eliminated by appropriate shielding, (ii) state the properties which an effective shield must have to eliminate the coupling yet not interfere with the TMS B field, and (iii) propose the mechanism for the long time scale (compared to the TMS pulse length time) of the observed interaction.

3) The Results contain the following statement: "…displacement current, if large enough, can stimulate neuronal structures through microelectrodes" and if the amount of such current injection is large enough, it will excite the neural tissues around the microelectrodes and severely confound the measurement of TMS effects." Adjacent to these sentences please provide estimates of what is "large enough" That is, how large would the current magnitudes have to be to introduce these potential problems?

4) Since the text mentions the problems with using a human-size TMS coil for rodent experimentation, then please elaborate on the expected size of the interactions in rodent sized coils.

5) Would the rebound excitation phase exist if a rat-sized TMS coil were used to stimulate the motor cortex rather than the human-sized coil used in this work? Please elaborate. This is not so much a criticism as it is a direction for further research (perhaps to be mentioned in the Discussion).

6) Please comment on whether the magnitudes of the interactions significantly influenced by the pulse type (i.e. monophasic or biphasic TMS pulse?).

7) Please consider commenting on this in the revised Discussion: While ML-oriented TMS evoked virtually no early response, PA-oriented TMS evoked neuronal spikes around 1-1.5 ms and at 3-3.5 ms. This finding is interesting and indicates the importance of coil orientation. However, since only a limited population of neurons was monitored, this study did not demonstrate that ML-oriented TMS would not have elicited early spikes in some other population of neurons than those that were monitored.

8) In the Discussion section, the reason for different neuronal responses to ML and PA stimulation is discussed but an essential aspect is forgotten: since the rat skull is far from spherical, the induced electric field does not remain the same (adjustment of the TMS intensity to keep the E-field constant was not mentioned in the text) if the coil is rotated; in the latter case, the E-field is likely to be much stronger than in the former case (ML). This fact alone might explain the fact that ML stimulation did not give rise to much early activity (but see the previous paragraph). In any case, in addition to this reviewer's "armchair thinking", please take another look at the ML and PA difference and explain the difference in E-field values (if any) in the ML and PA stimulations.

9) Please state the total number of recorded neurons in different animals and describe the methodology used for spike isolation and analysis. This is particularly relevant for TMS evoked response with stimulus orientation (where data are from only 4 animals).

10) Please respond to the following concern: The experiments related to different responses to TMS stimulus orientations is quite preliminary and it needs for additional data to be validated. Do the authors compare spike firing for both stimulus orientations maintaining the same isolated neurons? On the other hand, the authors claim that neurons were recorded at layer V of CFA. Do the authors test the putative nature (pyramidal, inhibitory interneurons…) of the recorded neurons? Could the orientation of the stimulus have a different impact on different neuronal populations located at layer V?

---

## [Author Response]

Essential revisions1) Organization and Presentation:a) Include, in the Introduction, a clear statement of the problematic system interactions, the manifestation of these interactions in the acquired EEP data and short statements about how each is to be ameliorated. This would include the interactions due to induction, electrical field (capacitive) coupling, vibration, and charge injection.

In the Introduction section of the manuscript, we added the following statements:

“The method is compatible with existing standard TMS coils and allows the recording of neuronal activities 0.8-1 ms after the onset of various types of TMS stimuli by attenuating artifacts resulting from magnetic induction, electric field coupling, and vibrations. Furthermore, the method allows for the instantaneous determination of TMS-driven inadvertent charge injection into the neural tissue, a problem that has been overlooked by almost all prior TMS-EEP studies. In the following sections, we will present…”

b) Presently the elucidation of the interactions between the TMS and EEP systems and their corrections come in three different places with increasing levels of detail: The Results, Materials and methods Section and the Supplemental Section. Please pare this down to two places with much less appearing in the supplemental section.

We improved the focus of the manuscript by moving information in the Supplemental Section to the main manuscript, and by condensing figures using *eLife*’s figure supplement option. The revised manuscript no longer contains a Supplemental Section and follows the outline listed below:

Introduction

1) Define the problem of TMS neurophysiology

2) Existing solutions and their limitations

3) A brief overview of our technical solution and of our initial physiological results

Results

1) Attenuation of the induction artifact

2) Attenuation of electric field coupling artifacts

3) Attenuation of vibration artifacts

4) Minimization and determination of inadvertent charge injection

5)In vivo method evaluation under various types of TMS stimuli

6) mspTMS evokes in the layer V of forelimb M1 a multiphasic rhythm of neuronal activities

7) mspTMS evoked short-latency (1-6 ms) neuronal responses differ with stimulus orientations

Discussion

1) A technical overview of our method

2) A discussion on results in the short-latency window and their links to human TMS.

3) A discussion on results in the long-latency window and their links to human TMS.

4) Proposed future directions for TMS-EEP.

And the figures are organized as indicated below:

Figure 1: Simultaneous TMS-EEP recording requires artifact attenuation in multiple dimensions. *(old Figure 1A)*

Figure 2: Simplified circuit diagram of the TMS-EEP amplifier. *(old Figure 1B Figure 1and Figure S1)*

Figure 3: Electric field coupling in TMS-EEP and its attenuation. *(old Figure 1C Figure 1and Figure S5)*

Figure 4: Low-noise miniature coaxial cable attenuates vibration artifacts. *(old Figure 1D Figure 1and Figure S6)*

Figure 5: TMS drives inadvertent charge injection in multiple loops formed by an EEP recording assembly. *(old Figure S3A-B; updated)*

Figure 5—figure supplement 1: The three-pronged electrode set design. *(old Figure S2)*

Figure 5—figure supplement 2: Circuit representations of the two induction loops shown in Figure 5. *(old Figure S3C-F)*

Figure 6: TMS-EEP recording setup and rapid signal recovery under the worst-case TMS stimuli. *(old Figure 2Figure 2)*

Figure 6—figure supplement 1: in vivo measurements of inadvertent charge injection. *(old Figure S4)*

Figure 7: mspTMS evoked multiphasic response alternating between excitation and inhibition. *(old Figure 3)Figure 3*

Figure 7—figure supplement 2: MUAP evoked by single-pulse ICMS. *(old Figure S7)*

Figure 7—figure supplement 3: Layer V neuronal response evoked by PA-oriented mspTMS at different intensities. *(old Figure S8)*

Figure 7—figure supplement 4: mspTMS evoked a multiphasic pattern of neuronal response in layer II/III. *(old Figure S9)*

Figure 7—figure supplement 1: Histological confirmation of electrode placement. *(old Figure S10)*

Figure 8: mspTMS-evoked short-latency neuronal responses differ with stimulus orientations. *(old Figure 4Figure 4)*

Figure 9: mspTMS activates different neuronal circuits depending on stimulus orientation or the time-window of investigation. *(old Figure 5)*

c) Clarify how the described methods differ from other methods that address these interactions. For instance, consider the following statement: "Two groups reported TMS-EEP methods for non-human primates (Mueller et al., 2014; Tischler et al., 2011); however, the applicability of these methods is very limited as they were developed solely for primate research, which is not best positioned to answer basic questions on neural circuitry, and they involve complex electronics and custom-designed TMS coils for artifact reduction." For this statement, please describe: (i) the methods that they used, (ii) the time window following the TMS pulse in which these methods are effective (0-1 ms as well?), (iii) how effective the methods are, and (iv) why the methods are not "best positioned to answer basic questions on neural circuitry."

To address these questions, in the Introduction section, we added the following statements:

“Recently, two groups reported TMS-EEP methods for non-human primate research. One of the methods utilized custom-built TMS coil and offline correction to minimize the TMS induced data loss to a median time of 2.5 ms (Tischler et al., 2011), while the other used a combination of custom-built coil, amplifier modifications, field sensing, active compensation, and offline correction to minimize the data loss to 1 ms (Mueller et al., 2014). Despite their success in artifact reduction, these methods face a major limitation that the technical expertise required for their implementations, especially custom-building TMS coils and field sensing, is not widely accessible to the neuroscience community. More importantly, these methods were developed solely for non-human primate research, which is used for investigating the neuronal underpinnings of high-level cognitive functions and therefore is not best suited for investigations concerning basic neurophysiology on the level of cells and detailed microcircuits.”

d) The manuscript content together with supplementary information has all the elements required for facilitating the proposed methodological improvements by interested researchers; nevertheless, much of this important information is categorized as "supplementary," giving priority to the observed physiological response to TMS (somehow incomplete data, see comments below). Taken into account that this manuscript has been submitted to "tools and resources" category all the important technical details should be included in the body of the manuscript.

Please see our response to Essential Revisions question 1b.

e) There are 5 "figures" in the main part and 10 in the supplement. However, in many figures, there are subfigures (A, B, C.…) and in some of the subfigures, there are multiple panels. The total number of labeled figures or subfigures is about 40. This may be too much. Even the main text contains 19 figures or subfigures. One reason for the large number of figures is that there is material in the manuscript for two manuscripts. For instance, while the abstract emphasizes the development of the engineering and measurement methodology, the discussion and much of the other text emphasize the neuroscientific results. Please consider condensing the text and figures to improve the paper's focus.

Please see our response to Essential Revisions question 1b.

2) The section describing the second artifact source which begins at the third paragraph of the Results section lacks clarity. This section can be improved if the authors: (i) state that capacitive coupling between the TMS and EEP systems poses a significant problem which can be eliminated by appropriate shielding, (ii) state the properties which an effective shield must have to eliminate the coupling yet not interfere with the TMS B field, and (iii) propose the mechanism for the long time scale (compared to the TMS pulse length time) of the observed interaction.

We restructured this section according to the comments above for improved clarity. Please see Results section.

3) The Results contain the following statement: "…displacement current, if large enough, can stimulate neuronal structures through microelectrodes" and "if the amount of such current injection is large enough, it will excite the neural tissues around the microelectrodes and severely confound the measurement of TMS effects." Adjacent to these sentences please provide estimates of what is "large enough" That is, how large would the current magnitudes have to be to introduce these potential problems?

It is the magnitude and the temporal pattern of such injected current that determine if the current injection is large enough to excite neuronal tissues and confound EEP measurements. To clarify this, we added the following statements:

“If the amount and the temporal structure of the injected current are comparable to the threshold parameters reported in intracortical microstimulation (ICMS) literature (bipolar charge transfer totaling from 150 to 800 pC, current waveform approximately similar to that of TMS; see Asanuma and Rosén, 1973, Butovas and Schwarz, 2003), such current will excite neuronal elements around the microelectrode tips and therefore severely confound the measurement of TMS effects.”

4) Since the text mentions the problems with using a human-size TMS coil for rodent experimentation, then please elaborate on the expected size of the interactions in rodent sized coils.

To address this question, in the Discussion section, we added the following statements:

“Nonetheless, the development of smaller and more compact coils specifically designed for small animals would be beneficial for their improved spatial resolution and smaller electromagnetic interference as the maximum magnetic output of these coils is much smaller (at mT level; Makowiecki et al., 2014; Tang et al., 2016) than the 4T output tested in our development.”

5) Would the rebound excitation phase exist if a rat-sized TMS coil were used to stimulate the motor cortex rather than the human-sized coil used in this work? Please elaborate. This is not so much a criticism as it is a direction for further research (perhaps to be mentioned in the Discussion).

To address this question, we added the following paragraph in the manuscript:

“Would the same neuronal activity pattern be observed if a rodent-sized TMS coil is used to stimulate the forelimb M1? We believe this is the case since we carefully calibrated coil position and stimulation strength according to MEP. Furthermore, the long-lasting inhibition and the rebound excitation are well-documented phenomena in ICMS (Butovas and Schwarz, 2003), which is a much more localized stimulation method than TMS. Additionally, as discussed above, data from human TMS is largely congruent with the pattern of neuronal activity reported here. However, we cannot rule out the possibility that the coil we used in this study directly activated structures outside of the forelimb M1. Nonetheless, the role of stimulus spatial resolution in modulating neural networks is a highly interesting topic for future research.”

6) Please comment on whether the magnitudes of the interactions significantly influenced by the pulse type (i.e. monophasic or biphasic TMS pulse?).

To address this question, we added the following paragraph in the manuscript:

“The amount of magnetic, electric, and vibrational interference TMS imposed on EEP depends on multiple factors. Some of the most critical factors include the waveform and magnitude of the pulsed magnetic and electric field emitted from a TMS coil, the size of circuit loops formed by an EEP recording assembly, and the coil position relative to these loops. Changes in these parameters will result in changes in the severity of different types of interference. For example, keeping the coil the same, by replacing a standard monophasic with a standard biphasic stimulator, coil-emitted fields will generate a longer period of magnetic and electric field interference due to the longer pulse waveform. However, the severity of interference might be lower if the coil and biphasic stimulator combination does not produce magnetic and electric outputs that are as high as those in the monophasic case. Similarly, miniaturization of TMS coils for small animals can also lead to a reduction in interference because of the reduced electromagnetic output of such devices. Furthermore, the integration of recording, reference, and ground electrode in one microfabricated electrode array can also reduce the severity of interference as such configuration significantly decreases the area of circuit loops exposed to TMS.”

7) Please consider commenting on this in the revised Discussion: While ML-oriented TMS evoked virtually no early response, PA-oriented TMS evoked neuronal spikes around 1-1.5 ms and at 3-3.5 ms. This finding is interesting and indicates the importance of coil orientation. However, since only a limited population of neurons was monitored, this study did not demonstrate that ML-oriented TMS would not have elicited early spikes in some other population of neurons than those that were monitored.

To address this concern, we added the following statements in the manuscript:

“It might be argued that the observed discrepancy in short-latency response is a result of bias in neuronal sampling. We believe this is rather unlikely, as short-latency spikes evoked by ML stimulation were absent across multiple recording sites within CFA (0 out of 7 sites) while the significant high-frequency spiking pattern was observed readily within CFA under PA stimulation (3 out of 4 sites). Additionally, in PA trials, the observed high-frequency spiking disappeared when we turned the stimulus orientation to ML. While we cannot rule out the possibility that mspTMS evoked early spike responses in areas other than the ones we monitored, our data supports the notion that in the layer V of CFA — the output layer of the rodent forelimb M1 — selectivity in stimulus orientation exists.”

8) In the Discussion section, the reason for different neuronal responses to ML and PA stimulation is discussed but an essential aspect is forgotten: since the rat skull is far from spherical, the induced electric field does not remain the same (adjustment of the TMS intensity to keep the E-field constant was not mentioned in the text) if the coil is rotated; in the latter case, the E-field is likely to be much stronger than in the former case (ML). This fact alone might explain the fact that ML stimulation did not give rise to much early activity (but see the previous paragraph). In any case, in addition to this reviewer's "armchair thinking", please take another look at the ML and PA difference and explain the difference in E-field values (if any) in the ML and PA stimulations.

To address this possible confounding issue, we added the following statements in the manuscript:

“Another confounding factor that might explain the discrepancy is the intensity difference of induced electric fields in the brain under ML and PA stimulation. Since the rodent skull is not spherical, with a given coil output, induced electric field in the ML direction (along the short axis of the skull) should be lower in intensity than that in the PA direction (along the long axis of the skull), raising the possibility that the observed high-frequency spiking pattern under PA stimulation is a result of high intensity of the induced electric field. However, motor thresholds under ML stimulation, in which induced electric field intensity is lower, were significantly lower than their PA counterparts (median_ML_= 61% MSO; median_PA_= 74% MSO; Wilcoxon rank-sum test, p=0.03). This is a strong indication that factors other than induced electric field intensity play a critical role in stimulus orientation selectivity. Therefore, we conclude that the observed response difference between ML and PA stimulation is unlikely to be caused solely by the difference in the intensity of induced electric fields.”

9) Please state the total number of recorded neurons in different animals and describe the methodology used for spike isolation and analysis. This is particularly relevant for TMS evoked response with stimulus orientation (where data are from only 4 animals).

To clarify the issue raised in this comment, in the Materials and methods section of the manuscript, we added the following statements:

“Spike isolation was performed using principal component analysis of the spike waveforms followed by a Gaussian mixture model with Kalman filters that track waveform drifts over time (Ecker et al., 2014). A total of 51 single units were isolated (L5_ML_ = 19; L5_PA_ = 14; L2/3_ML_=18); however, since at the present stage we are only interested in characterizing the response of M1 neuronal population to mspTMS, in the following analysis, we combine spikes from all single units as well as those that cannot be reliably isolated into a multiunit cluster.”

10) Please respond to the following concern: The experiments related to different responses to TMS stimulus orientations is quite preliminary and it needs for additional data to be validated. Do the authors compare spike firing for both stimulus orientations maintaining the same isolated neurons? On the other hand, the authors claim that neurons were recorded at layer V of CFA. Do the authors test the putative nature (pyramidal, inhibitory interneurons…) of the recorded neurons? Could the orientation of the stimulus have a different impact on different neuronal populations located at layer V?

Unfortunately, we did not test if the same neuron was held under different coil orientations. This can be difficult given that CFA L5 neurons remained largely unresponsive under ML stimulation. However, through a combination of ICMS and histology, we did confirm the recording locations were indeed in the L5 of CFA. Regarding the putative nature of the recorded neurons, we did not perform classification to explore this since it is beyond the scope of the present study, which is focused on method development and the first characterization of the response from L5 neuronal population in the forelimb M1 to mspTMS that evoked focal forelimb activations. Nonetheless, we do agree with the reviewers that it is plausible that stimulus orientation has a differential impact on different neuronal populations, and we believe this issue should be investigated further in future studies.